# Seismic Analysis of Historical Urban Walls: Application to the Volterra Case Study

**Giovanna Concu, Mariangela Deligia and Mauro Sassu \***

Department of Civil, Environmental Engineering and Architecture, University of Cagliari, 09123 Cagliari, Italy
* Correspondence: msassu@unica.it

**Abstract:** Several Italian cities are characterized by the presence of centuries-old historic walls, which have a cultural heritage value and, due to their structural role as a retaining wall, often influence the safety of adjacent buildings and infrastructure. Ancient urban walls are increasingly subject to instability and collapse phenomena, because the greater frequency of extreme meteoric events aggravates the static condition of the walls and of the wall–soil system. Since the seismic risk in the contexts in which the historical urban walls are located is often medium-high, it is advisable to evaluate the influence of soil moisture on the seismic response of the soil–structure system. In this paper, the seismic vulnerability of historical urban walls was examined through considering scenarios of both dry and wet soil, in order to evaluate the seismic response of the structure as a function of soil imbibition. Seismic vulnerability analyses were carried out on the case study of the historical urban masonry walls of Volterra (Italy), which have been affected by two major collapses in the last ten years. Seismic vulnerability was assessed by means of the limit equilibrium method and the finite element method, and through adopting proper soil imbibition models. The results highlight which sections of the walls are at greater seismic risk due to the presence of soil moisture, as well as the influence of soil imbibition on the structural safety and failure mechanism.

**Keywords:** historical urban walls; seismic vulnerability; soil imbibition effect; hydraulic vulnerability

## 1. Introduction

Very recently, on 6 September 2020, about 30 m of the Medieval Walls of Pistoia (Tuscany, Italy) collapsed. Unfortunately, this is just one of the many events that have affected historical urban walls in several Italian cities in the last ten years [1–3]. Only in Tuscany, famous all over the world for its cultural heritage, seven documented collapses have occurred [2], with a loss of about EUR 5,500,000, considering the repair costs only. These failures draw attention to the vulnerability of historical urban walls (HUW), often subject to poor maintenance and further threatened by the greater frequency of extreme meteoric events, which affect both the structure and the soil.

The preservation of building heritage such as HUW against natural hazards induced by climate change is a topical field of investigation [4–7]. HUW collapses are in general unexpected and involve portions of the surrounding area, in some case threatening the buildings and infrastructures nearby. Analysis of these failures highlights that the presence of moisture is a common denominator in many cases [2,8]. Water induced by flooding and rainfall threatens civil infrastructure [9–12], especially those placed in urban and sub-urban contexts, such as HUW and bridges of reduced span [12–16]. This evidence indicates the need to introduce the weakening effect of water into the failure mechanism. The presence of water in the soil can be from several sources, such as rainfall, groundwater, or artificial causes, including leaks in water pipelines or the management of white-water. The increasing and irregular rainfall induced by climate changes [17–20] is a threat both for new and for ancient structures.

The analysis of the behavior of masonry structures under the effect of the aforementioned phenomena, including earthquakes, is based on the implementation and use of the correct structural models. The selection of an appropriate model depends on the assumptions made about the walls. For instance, if the wall is regarded as rigid, typical nonlinear rocking analyses can be performed by considering the infrastructure as free-standing or restrained [21–26].

Geometry, masonry texture [27,28], and structural and material regularity [29–32] are significant factors affecting the static and seismic vulnerability of existing masonry. In some cases, these elements are more significant and affordable than tensile and compressive masonry strength and consequently box behavior [33–37]. Some approaches consider the elasticity of the wall and apply finite element models (FEM) [38] or simplified computing for the calculation of elastic stresses [39,40].

Recently, Zhang et al. [41] proposed a simplified probabilistic approach for seismic fragility analysis of masonry structures under a mainshock–aftershock sequence. The uncertainty of masonry structures and earthquake ground motions were considered, to generate a database of earthquake–structure samples. The comparison with FEM models showed a significant saving in computational time, while maintaining accuracy. Sansoni et al. [42] proposed an analytical procedure based on the simplified lateral mechanism analysis (SLaMA) method for seismic vulnerability assessment of unreinforced masonry (URM) structures, based on analysis at the member and subsystem levels. An experimental test from the literature was used as a benchmark to validate the method, and a 2D macro-mechanical FEM model was developed to extend the results and complement the validation. The results were in relatively good agreement with the experimental and FEM output. Isik et al. [43] carried out a structural analysis of five Turkish minarets, applying the macro-modeling technique and using a FEM model. The seismic behavior and weakness of the minarets were determined. Pandey and Khadka [44] performed a linear dynamic analysis to develop fragility curves for a URM building in mud mortar. The results highlighted that typical URM buildings in Dhulikhel (Nepal) are highly susceptible to out of plane failure. Malomo and DeJong [45] further enhanced and adapted a recently developed macro-distinct element model (M-DEM), to enable the modeling of the in-plain (IP)/out-of-plain (OOP) interactions and combined failure mechanisms of URM assemblies. To this end, a simplified modeling strategy, including the introduction of an additional vertical spring layer subdividing the bottom and top macro-blocks, as well as new corner block discretization and interface models to replicate interlocking strength at the intersection between orthogonal walls, was developed. Validation was carried out using previous test results, and a satisfactory agreement was found between the actual and predicted behaviors.

A strategic tool in collapse prevention is provided by modern remote monitoring strategies, such as satellite interferometry, that can be applied to monitor dams, bridges, and slopes. Hopper at al. [46] proposed a method based on persistent scatterer (PS) interferometric synthetic aperture radar (InSAR), which uses spatial correlation of interferogram phase to find pixels with a low-phase variance in all terrains, with or without buildings. The method was used to study the behavior of Volcán Alcedo, Galápagos. Crosetto et al. [47] described the deformation monitoring of the Vallcebre landslide (Eastern Pyrenees, Spain) using the differential interferometric synthetic aperture radar (DInSAR) technique and corner reflectors (CRs). Hasa et al. [48] proposed the application of the multi-temporal (MT) InSAR process to a series of radar images over the same region for infrastructures monitoring. Budillon and Schirinzi [49] showed the effectiveness of a recently developed synthetic aperture radar tomography (TomoSAR) technique in assessing both possible deformations and the thermal dilation evolution of man-made structures. The technique was tested in two case studies, concerning two urban structures in the city of Naples (Italy), using X-band SAR data. Milillo et al. [50] presented the first comprehensive multi-sensor cumulative deformation map of the Mosul Dam (Iraq) generated from space-based SAR

measurements from the Italian constellation COSMOSkyMed and the European Sentinel-1a satellite.

Advanced data analysis methods are also becoming strategic tools in monitoring the seismic response of structures and infrastructures. Łacny et al. [51] applied the probabilistic power spectral density, based on standard spectral density plots, to the seismic data collected over a long period from three seismic stations connected within the CERN Seismic Network. The analysis was used to observe and monitor the increase in ambient vibration levels over a long period during heightened heavy machinery work. Montisci and Porcu [52] proposed a neural-network-based tool for the early warning of ground settlement hazards in urban areas. Based on the analysis of MT-InSAR data through an unsupervised learning, the method found precursors of similar time-evolving phenomena. Zhao et al. [53] proposed a technique based on GIS data analysis to mine aftershock events in early aftershock sequences that are closely related to the mainshock fault and then, using these events, to quickly generate seismic intensity assessment maps. Martakis et al. [54] developed a Bayesian model updating (BMU) framework to leverage the modal data extracted from actual and instrumented URM buildings for seismic assessment. The structural response was measured at various levels of excitation amplitude prior to damage development, allowing for the evaluation of amplitude-dependence effect on the model-updating scheme and ultimately on the predicted seismic performance.

Analysis of the seismic response of large ancient masonry structures such as HUW requires that particular attention is given to the study of the soil–structure interaction (SSI). Lazizi and Tahghighi [55] applied modal, nonlinear static, and time-history analyses to evaluate the structural performance of Kashan Grand Bazaar (Iran), by considering two cases of fixed support and SSI. The results showed that SSI has a great influence on the dynamic properties, pushover capacity, failure mechanism, base shear, and displacement demands of a structure. Gunaydin et al. [56] assessed the structural performance of a historical masonry clock tower using both numerical and experimental processes, considering different types of SSI systems, identifying the numerical dynamic characteristics, model updating procedure, nonlinear time-history analysis, and the evaluation of seismic performance level. Cacciola et al. [57] addressed the seismic response of linear behaving structures resting on compliant soil through a novel application of the Preisach model of hysteresis for nonlinear SSI problems. The method was applied to the bell tower of the Messina Cathedral in Italy. Altunişik et al. [58] examined the effects of earthquake input models for different soil conditions on the seismic behavior of two historical buildings, Santa Maria Church of Trabzon (Turkey) and its Guesthouse Building, taking into account the earthquake input models and different soil conditions (hard, medium, and soft soil). The results showed that the SSI and the soil types significantly altered the structural responses of both buildings. Drougkas et al. [59] performed numerical modeling of a wall of the nave of Saint Jacob church in Leuven, subjected to differential settlement. The analyses included the monitoring of the settlement in the church over an extended period and SSI. The numerical results were compared with the in situ observed damage and with an analytical damage prediction model. Fathi et al. [60] investigated the seismic performance of a historical masonry building, Arg of Tabriz (Arge Alishah, NW of Iran). Static, modal, and nonlinear dynamic analyses were performed by considering both SSI and fixed base (ignoring SSI) cases. The results showed that SSI greatly affected the mode shapes and their frequencies and, depending on the frequency content of the records, could have an incremental or decremental effect on the structural responses. Piro et al. [61] compared analytical predictions based on the replacement oscillator approach for the results of 2D dynamic analyses of coupled soil–foundation–structure (SFS) interaction elastic models, varying the geotechnical and structural properties, such as the soil stratigraphy, foundation depth, and number of floors for single load bearing URM walls having either a shallow foundation or an underground floor embedded in layered soil.

However, only a limited number of studies have been devoted to SSI considering the effect of water content and soil imbibition, which is a crucial aspect, especially nowadays

due to the increasing frequency of extreme weather events. A pioneering study was carried out by Costantino in 1986 [62]. The study developed a FEM model for the two-phased formulation of the combined soil–water problem based on the Biot dynamic equations of motion for both the solid and fluid phases of a typical soil in the case of a typical nuclear reactor building. The results showed that the interaction coefficients were significantly modified as compared to a dry soil, particularly for the rocking response mode. More recently, Sáez et al. [63] investigated the influence of inelastic dynamic SSI (DSSI) on the response of moment-resisting frame buildings. Sandy soil in both dry and fully saturated conditions was analyzed, revealing that the influence of DSSI on dry soil was highly erratic; however, it tended to be invariantly favorable or negligible when the soil was in a saturated condition. Liratzakis and Tsompanakis [64] examined the dynamic response of an ordinary stone URM, considering the SSI along with the nonlinear behavior of both soil and structure. Analyses were repeated for eight different saturation levels covering a wide range of soil conditions. The results showed greater drifts when the structure was constructed on relatively dry soil. Lalicata et al. [65] explored the effects of partial saturation of soil on the response of a single pile subjected to a combination of lateral force and bending moment under drained conditions. The results showed a marked influence of soil partial saturation on the pile response, under both working loads and ultimate loads. Vagaggini et al. [11] developed an analytical model for a retaining HUW interacting with soil, whose mechanical properties are influenced by the soil penetrating moisture due to rainfall.

To the best of authors' knowledge, only studies [11] and [64] concerned masonry structures, with [11] being specific to HUWs, but without a seismic vulnerability assessment.

Since the assessment of the role of soil imbibition in the seismic response of large masonry buildings, such as HUW, is still a relatively unexplored issue, the present paper aimed to contribute to filling this gap, given the importance of the topic in light of the current climate emergency. In this paper, the seismic vulnerability of a HUW is examined, considering scenarios of both dry and wet soil conditions and adopting a proper imbibition model for the soil. A seismic vulnerability analysis was carried out on the case study of the HUW of Volterra (Italy), which has been affected by two major collapses in the last ten years. Seismic vulnerability was assessed by means of the limit equilibrium method (LEM) and the finite element method (FEM), and both nonlinear static (NLS) and nonlinear dynamic (NLD) analyses were carried out. In detail, the study was developed according to the following scheme:

(1) analytical assessment of seismic vulnerability considering different scenarios of soil saturation by adopting proper soil imbibition models,
(2) seismic NLS analysis using LEM,
(3) seismic NLD analysis using FEM,
(4) results comparison and discussion.

The results demonstrate the effect of soil moisture on HUW structural safety.

## 2. Case Study

The ancient wall perimeter of Volterra in the III century B.C. had a total extension of about 7.3 km, enclosing an area of approximately 116 hectares and representing the most important Etruscan fortress. Nowadays, the extension of the Volterra's HUW is reduced to the historic center only (Figure 1) and preserves the original beauty, as well as the early plane–altimetric complexity. According to the seismic classification of the Italian territory, Volterra belongs to seismic zone 3; that is, an area with low seismic hazard that can be subject to modest shaking, in which rare strong earthquakes can occur. The horizontal acceleration (ag) with a probability of being exceeded equal to 10% in 50 years is included in the range $0.05 < ag \leq 0.15$.

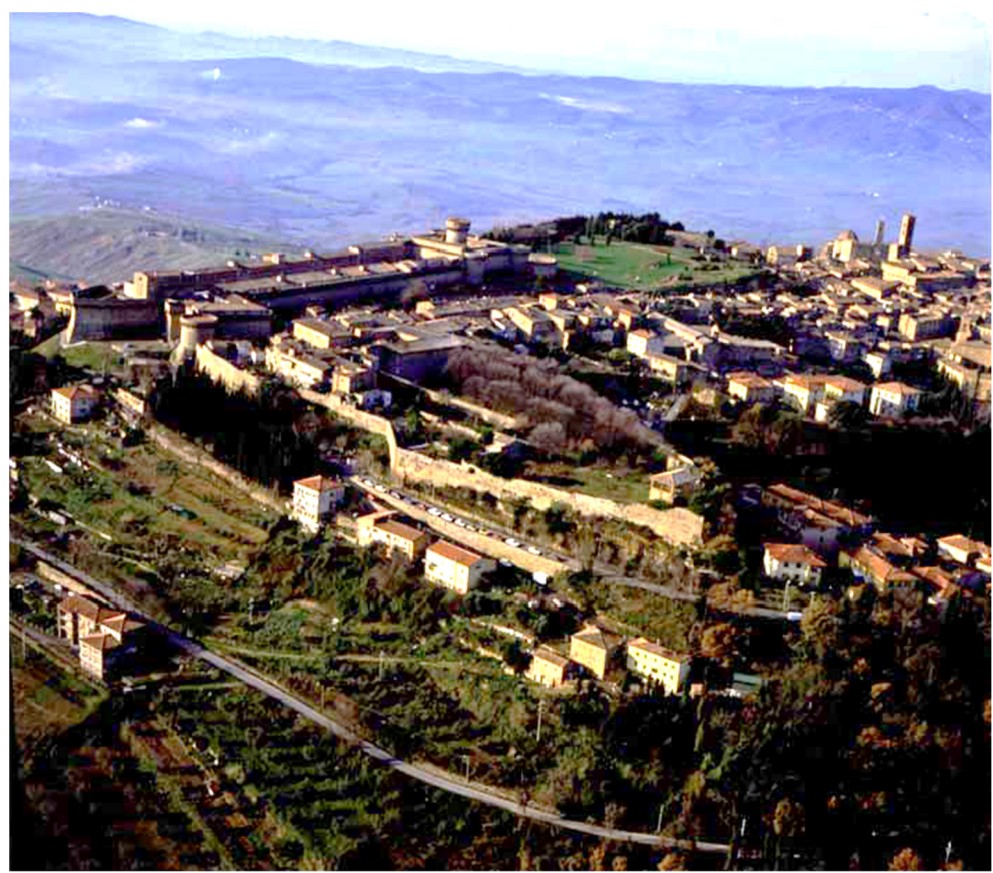

**Figure 1.** Aerial view of the Volterra urban walls and the fortress [35].

Historical building heritage requires a high level of attention and specific strategies of surveillance, and vulnerability investigations and maintenance should be developed. When dealing with historical walls, it is essential to characterize the sections to be analyzed, in order to identify the most vulnerable. Modern technology furnishes sharp and efficient survey methods. A proposal for a large-scale survey for a HUW was reported in [7,12]. On this basis, the urban perimeter was divided into several sections that were progressively numbered, as shown in Figure 2. The most representative sections, considering the most recurrent but also the most critical from a technical point of view, are listed in Table 1. These sections were classified with respect to slenderness $\lambda$ (Equation (1)) and filling ratio $\varphi$ (Equation (2)), the filling ratio being lower than one.

$$\lambda = h/b \tag{1}$$

$$\varphi = \frac{h_s}{h_w} \tag{2}$$

where

$b$    average thickness of the wall.
$h_w$ height of the wall, including the presence of railings.
$h_s$    height of the backfill soil.

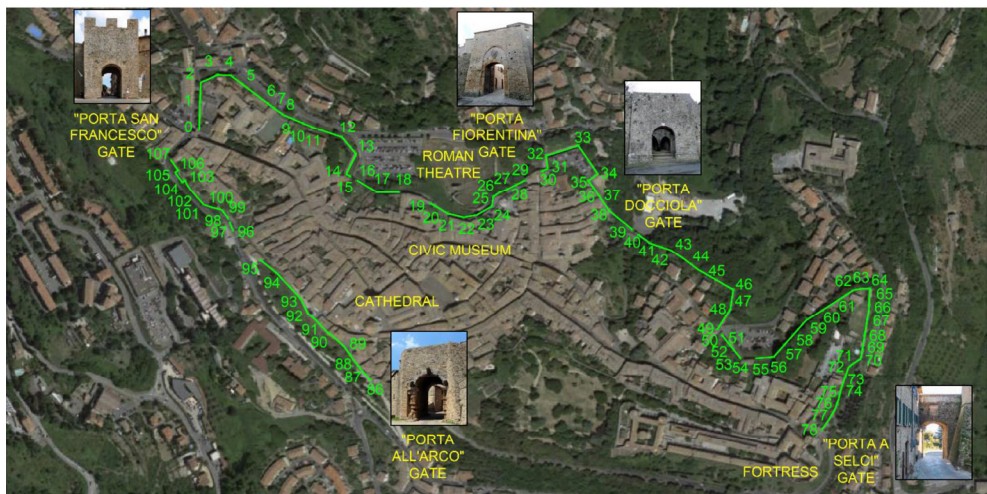

**Figure 2.** Perimeter of medieval walls of Volterra divided into numbered sections.

**Table 1.** Volterra's HUW analyzed sections. Geometry, slenderness λ, and filling ratio φ. Dimensions are in meters.

| Section | Sketch | Pictures |
|---------|--------|----------|
| 32–33<br>λ = 3.81<br>φ= 0.90 | | |
| 48–49<br>λ = 3.78<br>φ= 0.94 | | |

**Table 1.** *Cont.*

| Section | Sketch | Pictures |
|---|---|---|
| 50<br>$\lambda = 2.85$<br>$\varphi = 0.00$ | | |
| 61–62<br>$\lambda = 3.52$<br>$\varphi = 0.15$ | | |
| 71–72<br>$\lambda = 3.09$<br>$\varphi = 0.69$ | | |

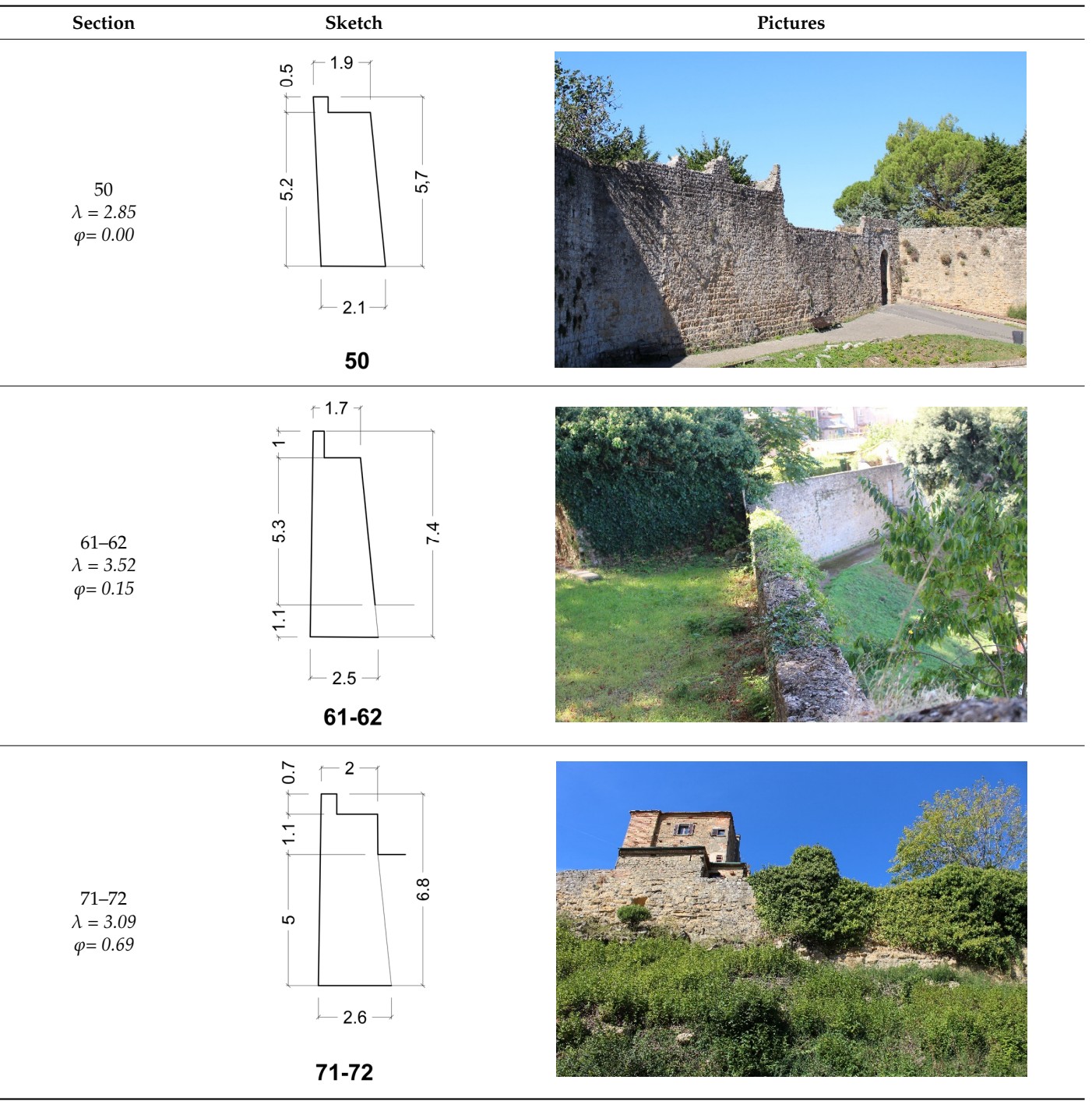

**Table 1.** *Cont.*

| Section | Sketch | Pictures |
|---|---|---|
| 87 <br> λ = 3.55 <br> φ= 0.91 | | |
| 92–93 <br> λ = 3.78 <br> φ= 0.9 | | |

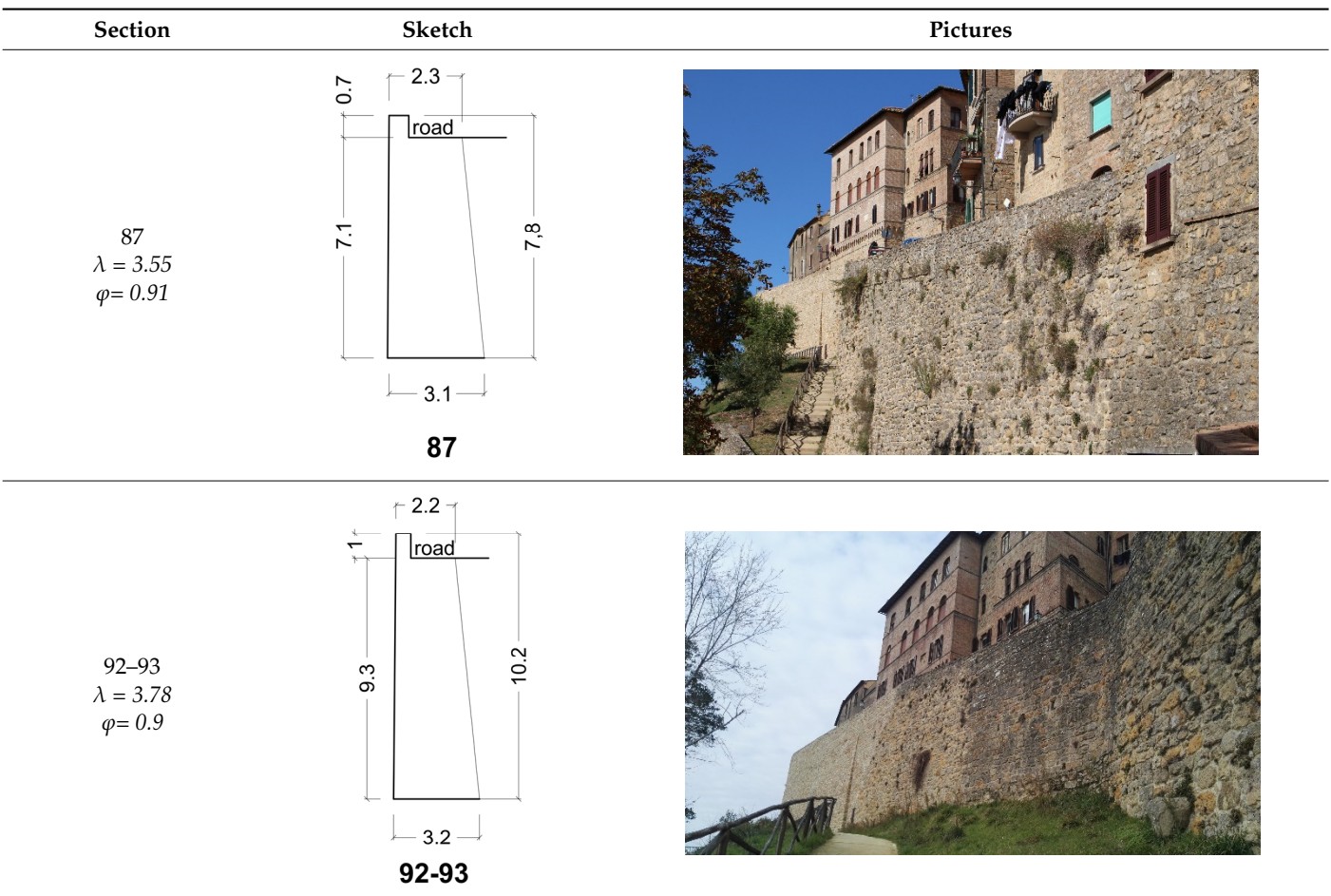

## 3. Analytical Assessments of Seismic Vulnerability

The study of a HUW requires defining some specific performance indicators, as occurs in the seismic design of buildings. Buildings and retaining structures present several states towards collapse. Four specific limit states (LS) were proposed in [40] for HUW: (1) collapse (SLC), (2) lifeguard (SLV), (3) damage (SLD), and (4) integrity (SLI). In case of the situation of a retaining wall, the relative displacement $d_r$ (Figure 3a) is defined by Equation (3):

$$d_r = d_T - d_B \tag{3}$$

where $d_T$ is the displacement of the top and $d_B$ of the basement. In Figure 3b, three representative sections in which to carry out seismic assessment are highlighted, one in the middle of the wall (section 1) and two at the basement of the wall (sections 2 and 3). Section 2 is taken in correspondence with the masonry, whereas section 3 is taken in correspondence with the soil.

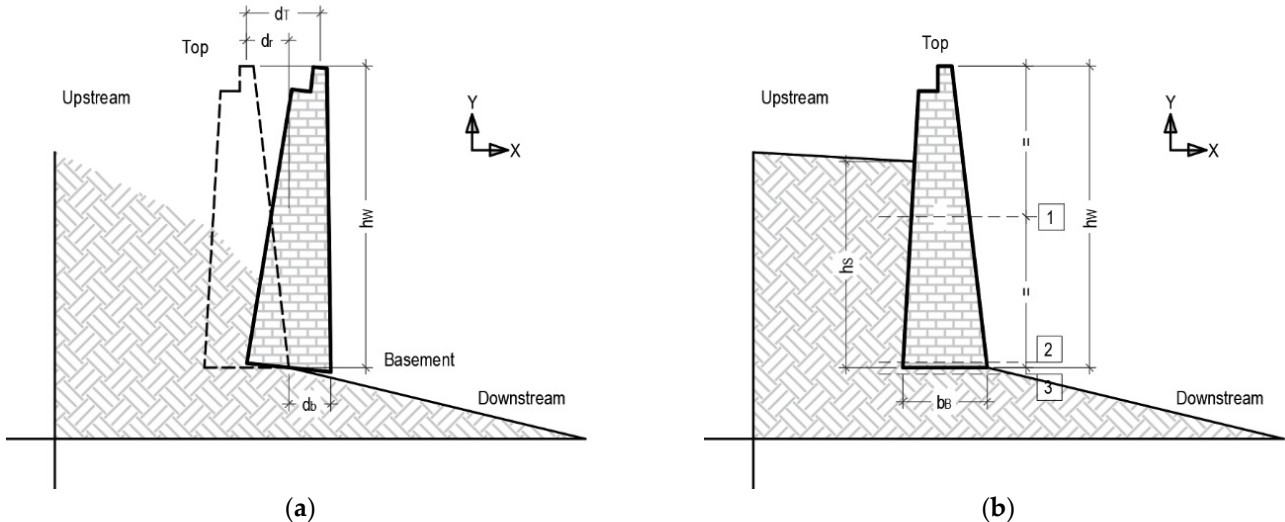

**Figure 3.** Typical drawing of a HUW. Relative displacements (**a**) and verification sections (**b**).

Considering the structural typology and the type of analysis (plane analysis), the relative displacement is assumed as the controlling parameter in the identification of LS (Table 2). For SLI, expert judgement is considered as a control parameter for the integrity, whereas the collapse displacement $d_{SLCg}$ is evaluated case by case through a nonlinear static analysis. Lifeguard displacement $d_{SLV}$ is assumed to be equal to 90% of collapse displacement $d_{SLCg}$, and damage displacement $d_{SLD}$ is assumed as a fraction of the height of the wall $h_w$.

**Table 2.** Limit states in [40].

| $\mathbf{d_{SLCg}}$ | $\mathbf{d_{SLV}}$ | $\mathbf{d_{SLD}}$ | $\mathbf{d_{SLI}}$ |
|---|---|---|---|
| Incipient collapse | $0.9 \times d_{SLCg}$ | $h_w/100$ | Expert judgement |

*3.1. Moisture Effect*

In general, the accumulation of water from several sources induces a double effect: (1) increase of the loads, especially the horizontal trust; and (2) reduction of soil mechanical properties. On that basis, a model of imbibition considering the weakening effect induced by the presence of moisture is proposed.

The reduction of soil mechanical properties was analyzed in the case of indefinite slopes (SLIP Model), as proposed by Montrasio and Valentino in [66,67]. In particular, the SLIP Model considers the reduction of cohesion with respect to the soil saturation. In addition, the results of Yoshida et al. [68] considered the effect of saturation on shear strength.

Let us consider a discretization of the soil in horizontal strata of height $H$ (Figure 4). The portion of height $mH$ ($m < 1$) is considered fully saturated, with the complementay part being $(1 - m)H$ partially saturated.

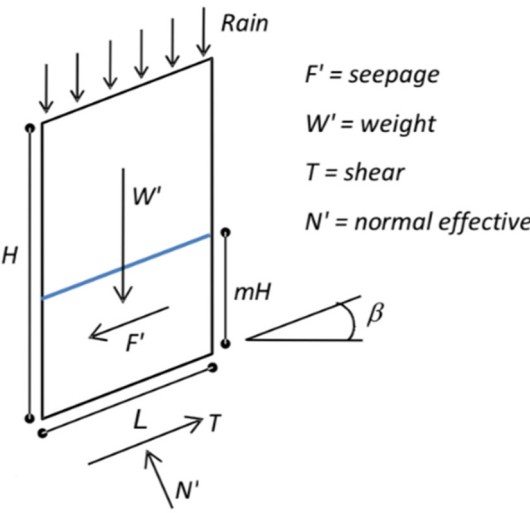

**Figure 4.** View of the imbibition level for the slip model in [66,67].

Each stratum is characterized by a uniform percentage of saturation expressed as:

$$m = \frac{\beta^* h}{n(1 - S_r)H} \tag{4}$$

where

$h$　height of rainfall
$H$　height of the soil interested by the rainfall
$\beta^*$　capacity of imbibition or percentage of filtered rain
$n$　soil porosity
$S_r$　saturation grade.

$H$ depends on the presence of impermeable soil layer. $\beta^*$ can be estimated [48] as 70%. The grade of saturation is expressed by Montrasio et al. [10] as:

$$S_r(h) = S_{r0} + \frac{\beta h}{nH} \tag{5}$$

where

$S_{r0}$　initial saturation grade
$\beta$　capacity of imbibition
$h$　height of rain.

The initial saturation grade $S_{r0}$ obviously depends on the soil moisture and is related to the weather of the previous days. In this study an initial saturation grade $S_{r0}$ = 0, 30 is assumed, in accordance with recurrent moisture scenarios at the site of Volterra.

Considering the pluviometry of the site and using Equation (4), it is possible to relate a generic saturation grade $S_r$ to the return period of the event that caused it. In this case this aspect is overlooked because the moisture scenarios are not necessarily related to rainfall.

In the case of non-saturated soil, the shear resistance is expressed through a modified Mohr-Coulomb law:

$$\tau = c' + \sigma' \tan \phi' + c_\psi \tag{6}$$

where

$c'_\psi$　initial apparent cohesion (Fredlund and Rahardjo) [69].

The initial apparent cohesion $c'_\psi$ can be expressed as:

$$c'_\psi = A S_r (1 - S_r)^\lambda \tag{7}$$

where $A$ and $\lambda$ are dimensionless coefficients that depend on the soil type and are identified by experimental tests from Montrasio and Valentino [66,67,70,71]. For the sake of simplicity, it is possible to apply an average value of $c_\psi$ that can be applied to the entire depth $H$:

$$c_\psi = c'_\psi (1 - m)^\alpha \tag{8}$$

where

$\alpha$　　homogenization coefficient, here assumed as equal to 3, 40 [30].

In this study, the analysis of masonry sections was carried out considering a double soil scenario: (a) dry conditions, and (b) partially saturated conditions. In the second, a portion of soil characterized by a depth of 1.00 m was considered to be in a partially saturated condition. It is worth noting that earthquake analyses are usually only carried out with dry soil, and usually in non-draining conditions. This study aimed to only evaluate the seismic behavior of the walls through a numerical investigation with different soil hydrological configurations.

*3.2. Seismic Analysis*

The seismic vulnerability of Volterra's HUW was evaluated with the limit equilibrium method (LEM) and with the finite element method (FEM).

In both cases, plane models were implemented, referring to the more representative sections of the urban perimeter, as defined in Table 1.

First, an LEM analysis was performed using the freeware software SSAP 2000 [72], to model the strata that characterize the sections and also considering the role of imbibition. The seismic load was characterized through a pseudo-static approach [73], and Sarma's method [72,74] was used to calculate the minimum safety factor (SF). The results are presented as a contour map of SF (Figure 5b), showing the relative failure surfaces (Figure 5a).

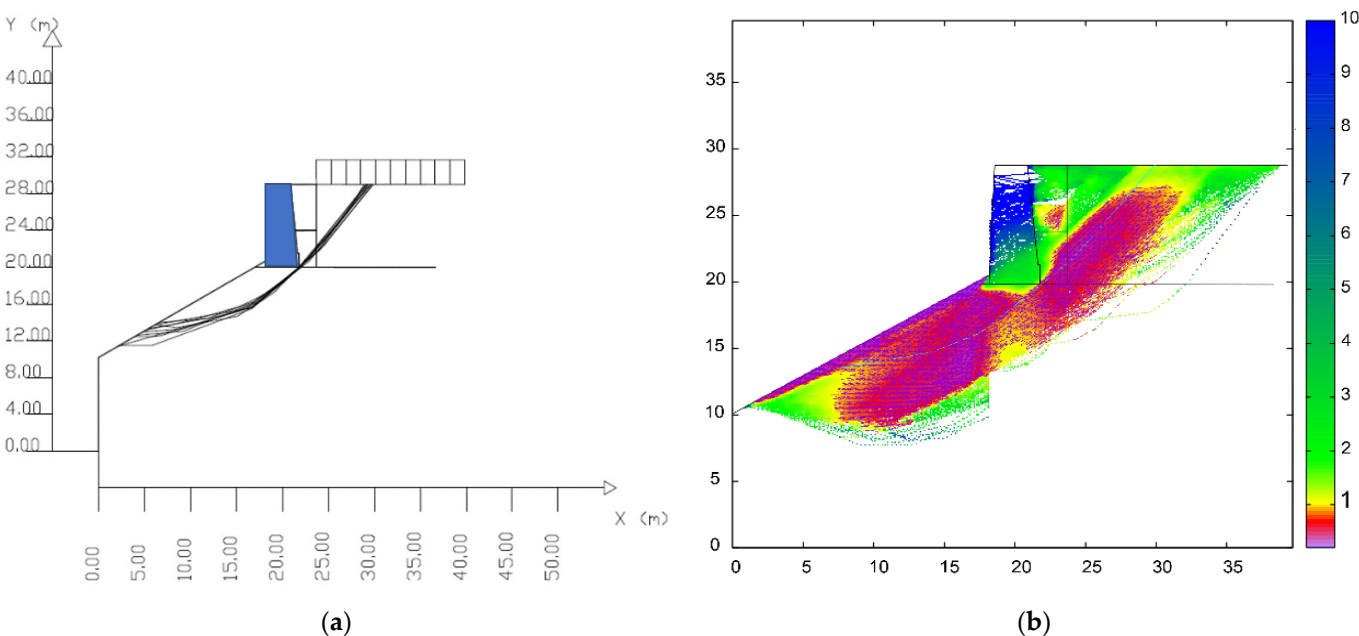

(**a**)　　　　　　　　　　　　　　　　　　　　　　　　　　　　　　(**b**)

**Figure 5.** LEM analysis of sections 92–93 (Table 1) with the software SSAP2000 [73]. Surface failure (**a**) and contour of the safety factors (**b**).

The contour map of the representative sections shows wide areas characterized by SF < 1 (red and magenta in Figure 5b). This earthquake vulnerability is highlighted in several sections characterized by a high filling ratio.

The same representative sections were also modelled with the FEM Software Straus 7 [75]. The mesh was formed by eight node–plate elements of dimensions between 0.40 m and 0.75 m. For the masonry and soil, two types of plate element, *Quad8-isotropic* and *Quad8-soil*, were chosen. The second type is specific to soil modelling and allows taking into account the initial stress induced by the consolidation effect.

Figure 6 shows the model of the representative sections 92–93 in two soil configurations: (Figure 6a) with, and (Figure 6b) without imbibition.

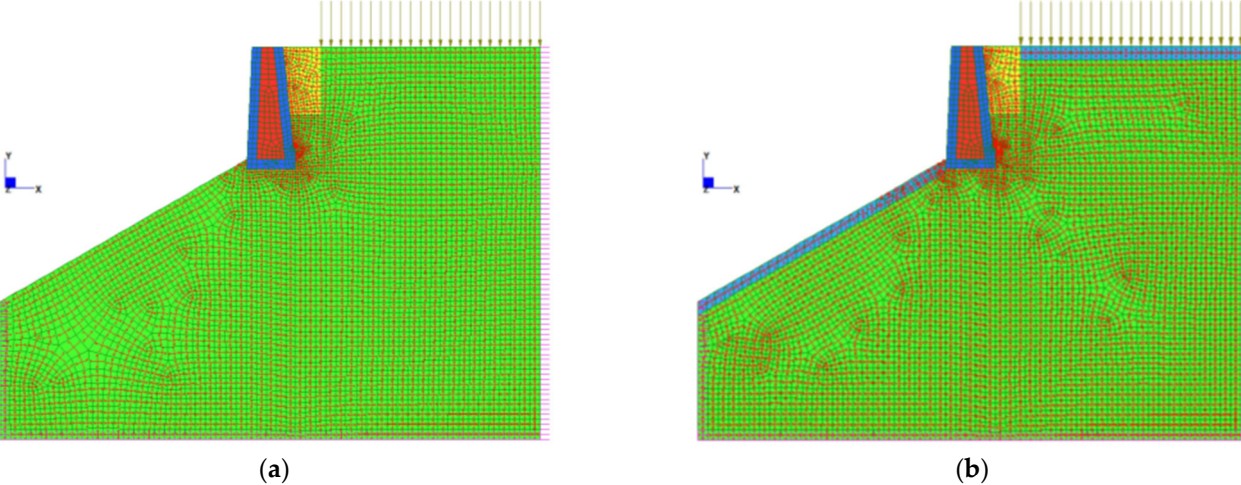

(**a**)                               (**b**)

**Figure 6.** FEM model of sections 92–93. On the left, the standard model (**a**); and on the right, the modified model (**b**) that considered imbibition.

Similar FEM models were realized for the other sections, considering a significant earth volume. With $h_w$ being the height of the wall, a horizontal dimension of 4 $h_w$ and a depth of 2 $h_w$ under the wall foundation were considered.

The displacements were fixed in both x and in y directions for the base of the model, and only in the X direction on the lateral side. The potential presence of buildings was taken into account using an equivalent load and mass.

For each section, two types of model were implemented: (a) standard model, and (b) modified model that considered imbibition (Table 3).

**Table 3.** Implemented models.

| Model | |
|---|---|
| (a) standard | (b) with imbibition |
| Uniform soil in dry conditions. | Stratum of saturated soil with a depth equal to 1.00 m and uniform soil |
| Foot print S | Foot print I |

The soil is mainly made up of alternate limestone and sandstone formations. It is weakened by rain and water filtration, as described in [40]. The soil mechanical properties were based on site coring and other in situ tests documented after the collapse of 2014 [7,48]. The masonry is made from a local soft stone called "Pietra Panchina", typical of the Volterra area. The recurrent section is characterized by a double external face/stratum with a regular stone texture, with a poorer quality infill material inside. The external strata are not connected, except for the top and the base of the wall.

The determination of the seismic load in the case of a retaining wall must consider the participating mass of the adjacent soil. The following procedure was performed: (1) modal analysis with evaluation of natural frequencies with the major mass participation in the

horizontal (x) and vertical (y) directions (Figure 7); (2) determination of the spectral acceleration of the respective components; (3) application of relative pseudo-static acceleration to the models. This allowed a more accurate application of static loads rather than imposing peak accelerations on the ground as loads.

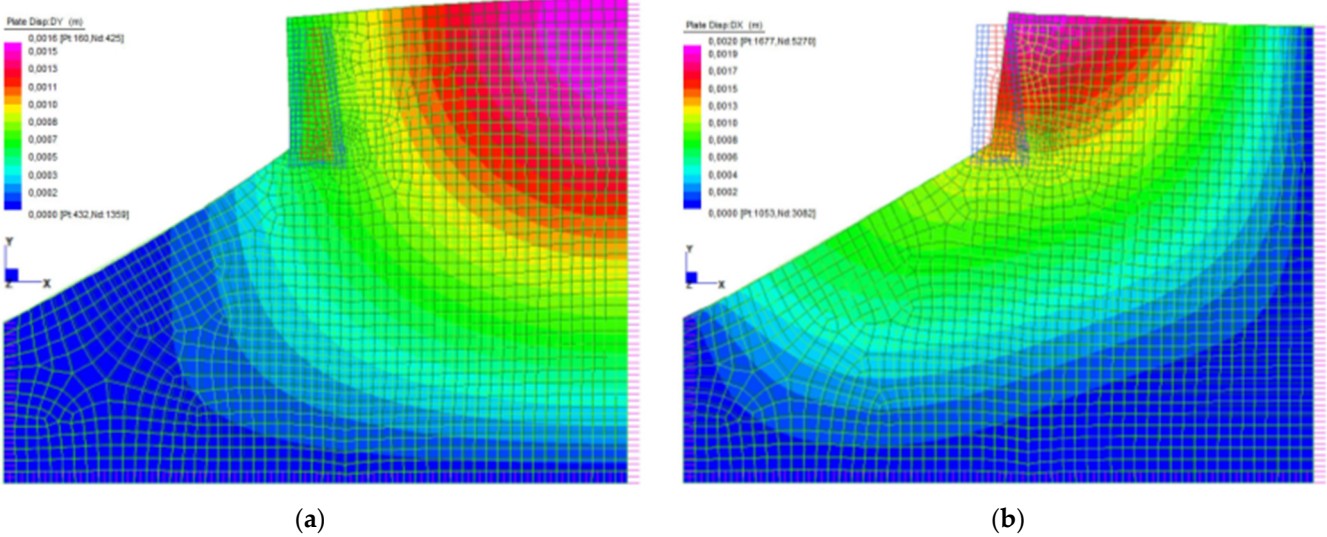

(**a**)                                                                                                                  (**b**)

**Figure 7.**   Sections 92 and 93, (**a**) second mode—participating mass 40.9%; (**b**) first mode—participating mass 54.4%.

NLS was carried out by first applying the effect of vertical loads such as gravity and external loads induced by adjacent buildings, then introducing the seismic load. Both vertical and horizontal seismic loads were combined, as shown in Table 4.

**Table 4.** Seismic load combinations used in NLS.

|    | Combination 1 | Combination 2 | Combination 3 | Combination 4 |
|----|---------------|---------------|---------------|---------------|
| ax | +ax | +ax | −ax | −ax |
| ay | +ay | −ay | +ay | −ay |

The performance point that represents the intersection between the equivalent bilateral curve and the response spectrum in the ADSR plan was calculated with N2 Method [47,48] for each analyzed section.

For each section, the resistance, equilibrium, and load-bearing capacity were calculated, highlighting that in some sections a collapse would take place before reaching the performance point. Moreover, tensional checks were made during the steps of the nonlinear static analysis in two relevant sections, one located at the top of the wall and another at the base.

## 4. Results

NLS analyses were performed for the sections listed in Table 1. A comparison of the capacity curves is illustrated in Figure 8, along with the load factor that led to collapse.

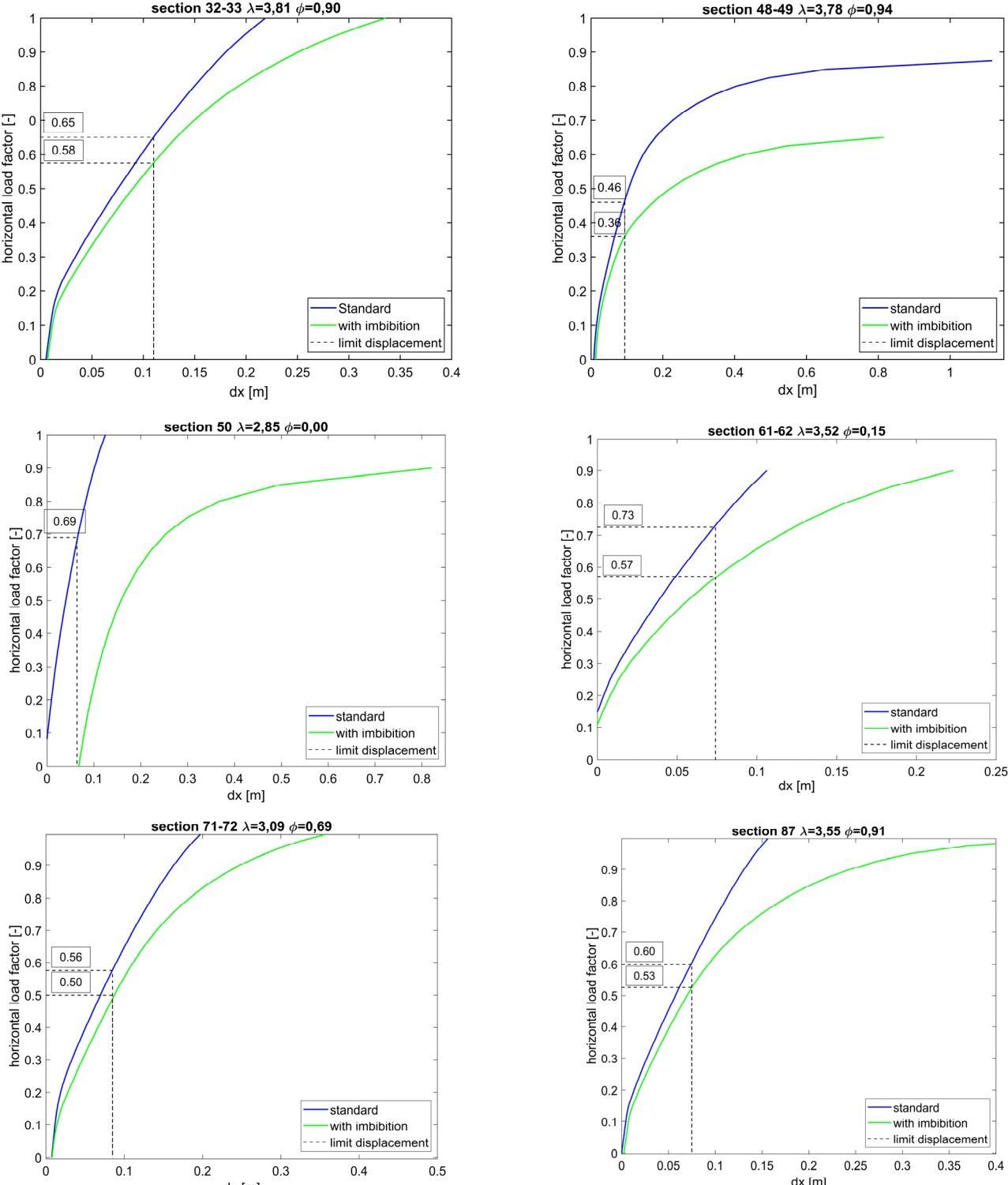

**Figure 8.** *Cont.*

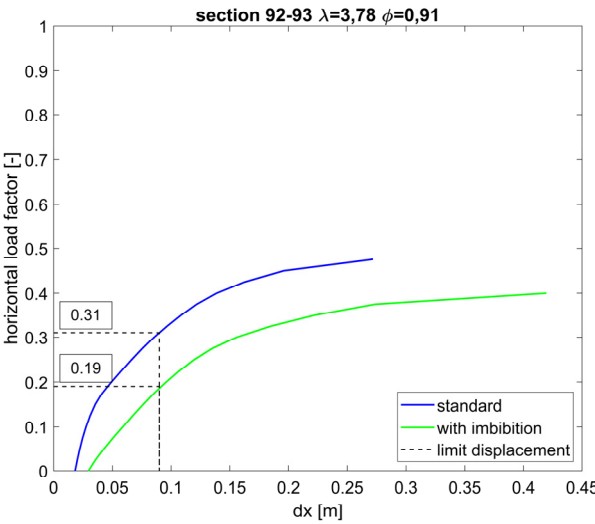

**Figure 8.** Push over curves for the examined sections.

The ultimate displacement in the case of imbibition ($\mu_i$) can be related to the ultimate displacement in the case of a dry section ($\mu_d$) using Equation (9), with $\Delta\mu$ being the displacement ratio at failure. The results are listed in Table 5 and show that $\mu_i$ was generally higher than $\mu_d$.

$$\Delta\mu = \frac{\mu_i - \mu_d}{\mu_d} \tag{9}$$

**Table 5.** Slenderness, filling ratio, and displacement ratio at failure.

| Section | λ | φ | Δμ |
|---|---|---|---|
| 32–33 | 3.8 | 0.9 | 0.50 |
| 48–49 | 3.8 | 0.9 | −0.23 |
| 50 | 2.9 | 0.0 | 5.83 |
| 61–61 | 3.5 | 0.1 | 1.00 |
| 71–72 | 3.1 | 0.7 | 0.75 |
| 87 | 3.5 | 0.9 | 1.50 |
| 92–93 | 3.8 | 0.9 | 0.56 |

Figure 8 shows that the ultimate capacity generally decreased with imbibition. Considering the imbibition in the seismic analysis of the section led to greater displacements than in the dry section, with the same increase in the horizontal load. This result increased the importance of considering the scenarios with imbibition. Furthermore, it should be noted that the effect of imbibition was different according to the different boundary conditions, such as the different initial displacements in the attribution of the initial equilibrium conditions resulting in a different ultimate ductility. The change in shape and scale of the pushover curves according to the section depended on boundary conditions and type of failure. The different types of failure, considering the two scenarios (a) with and (b) without imbibition, are shown in Figure 9, in which the seismic load that leads to collapse is expressed in terms of the corresponding return period.

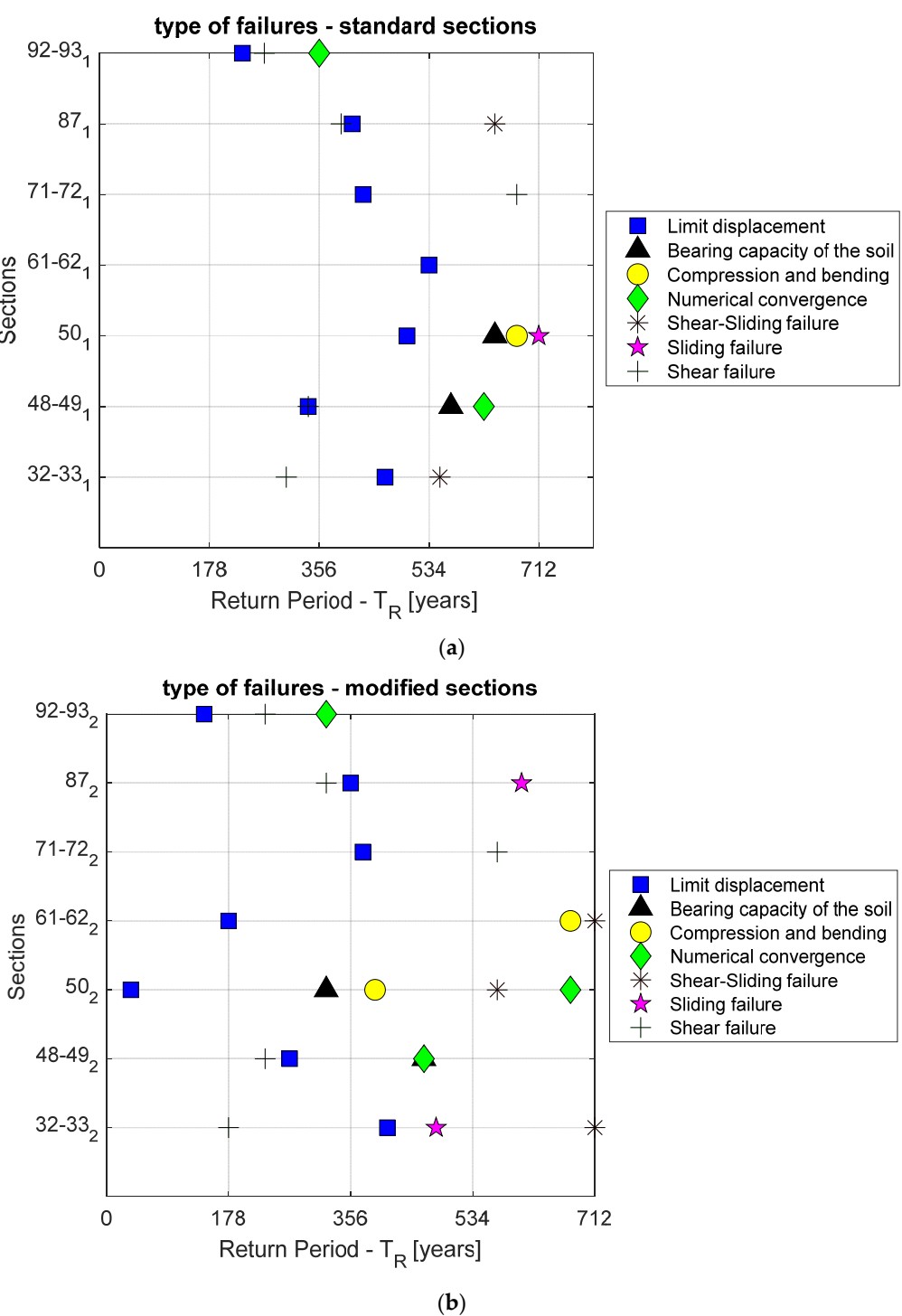

**Figure 9.** Types of failure for the analyzed section (NLS analysis). Standard (without imbibition) section (**a**) and modified (with imbibition) section (**b**).

The collapse of a section was detected by conventional masonry or soil failure mechanisms (compression–bending, shear, shear–sliding, sliding, bearing capacity of the soil) or, in some cases, using the numerical convergence of the FEM model. In the NLS analysis, the load increments were gradually reduced to find the most probable collapse type. For some sections (48–49, 50, 92–93) the displacement diverged, denoting the overturning of the masonry walls.

A comparison of the failure surfaces for sections 48–49 and 92–93 is presented in Figure 10. This figure shows that the considered sections collapsed in different modes. The limitation of the displacement (blue square) can be assumed as a prudential indicator of the failure, as well as of the damage. Slight differences between the sliding surfaces obtained from different types of analysis are evident. Section 92–93, where the slope is greater, exhibits greater differences than section 48–49.

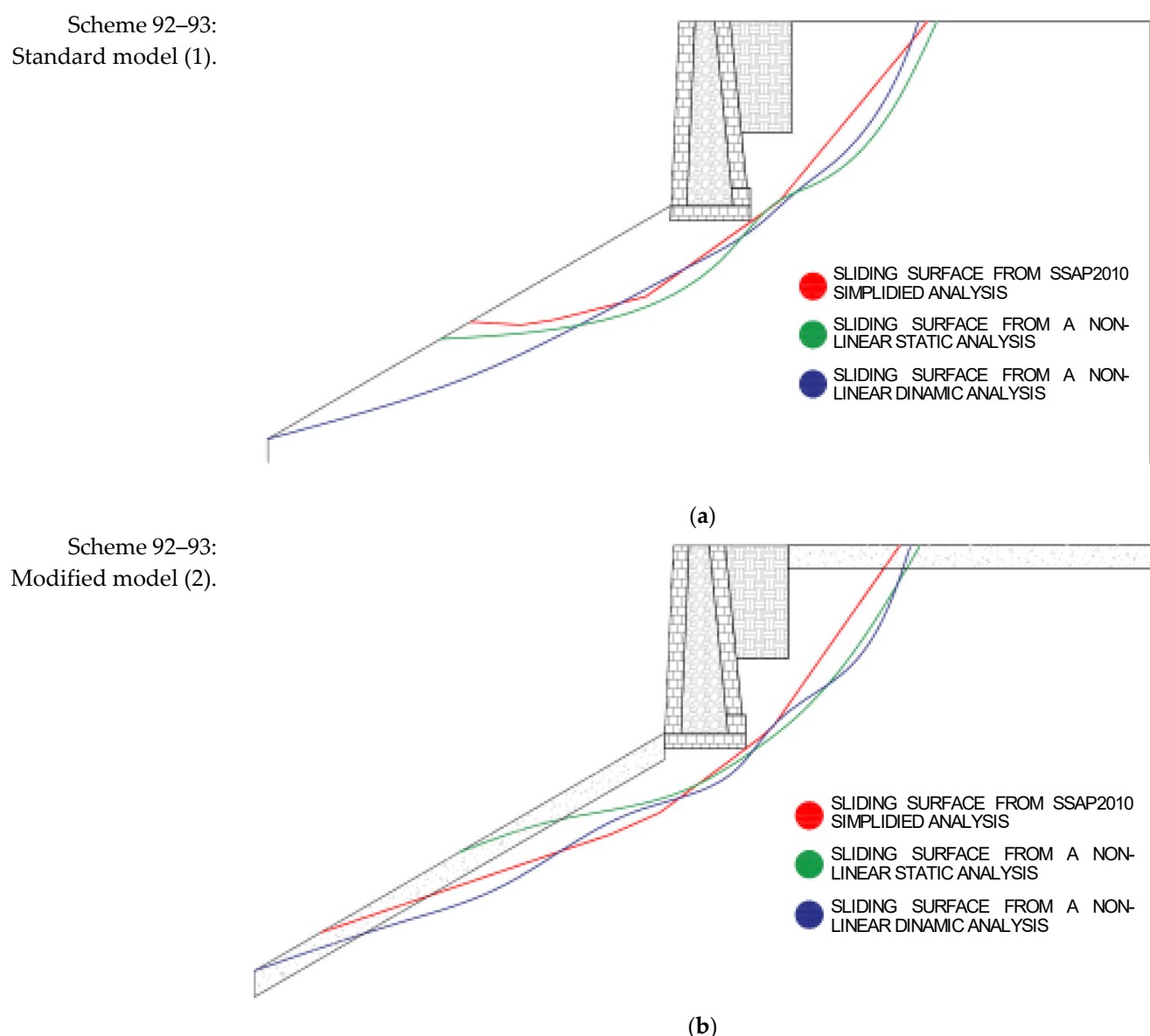

**Figure 10.** *Cont.*

Scheme 48–49:
Standard model (1).

Scheme 48–49:
Modified model (2).

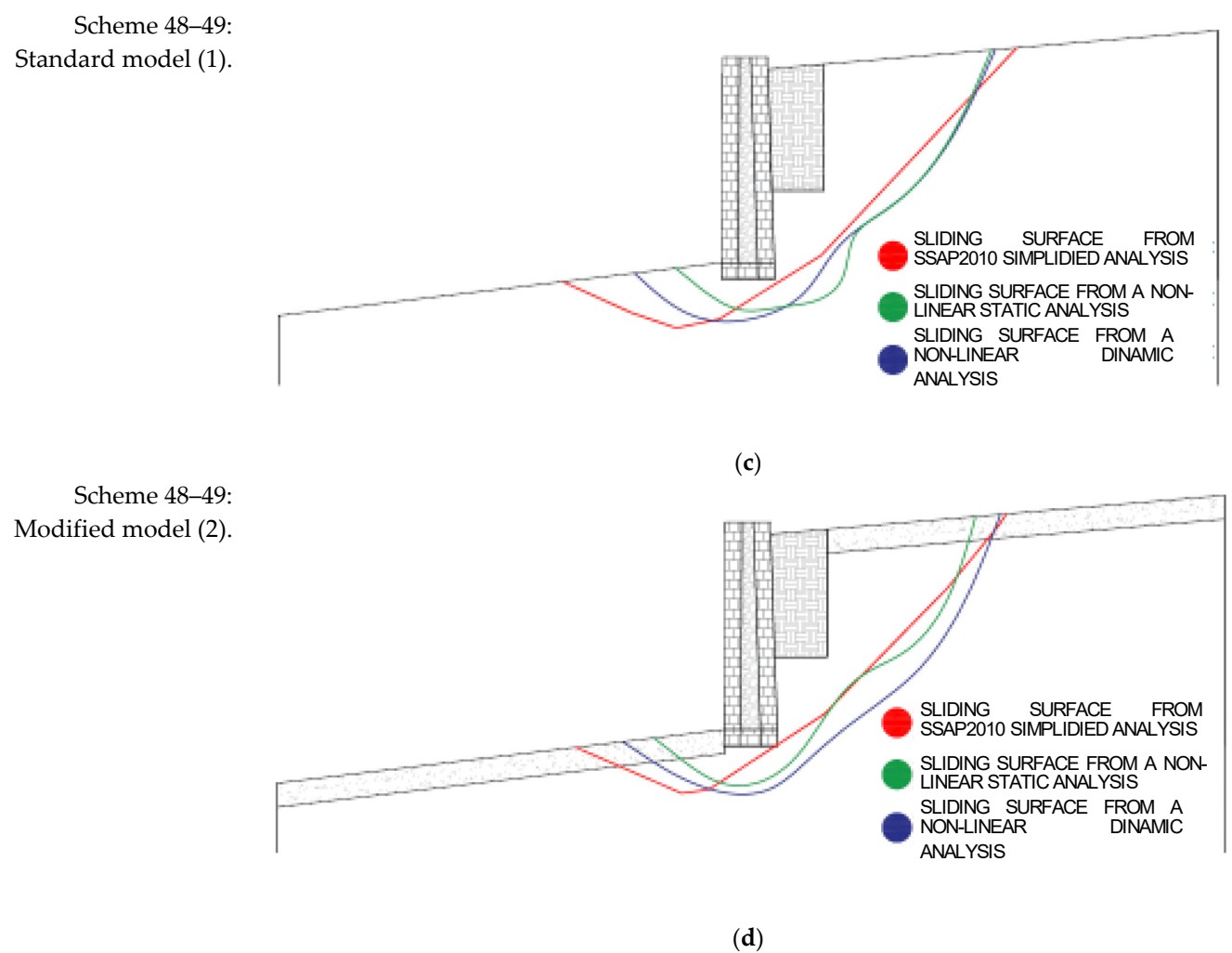

**Figure 10.** Failure surface of two representative sections for the standard (**a**,**c**) and modified sections (**b**,**d**).

Figures 11 and 12 show the demand/capacity (D/C) ratio, respectively, in the case of compression–bending failure and overturning for the considered sections. The regression was performed considering data from the both NLS analysis and NLD analysis and was evaluated as a function of the filling ratio φ (Figures 11a and 12a) and slenderness λ (Figures 11b and 12b). It can be noted that most of the sections had insufficient capacity with respect to the seismic demand. In particular, the sections characterized by a higher level of filling were the most vulnerable. Compression–bending failure was the most probable collapse mechanism and the one most correlated with the filling ratio and slenderness.

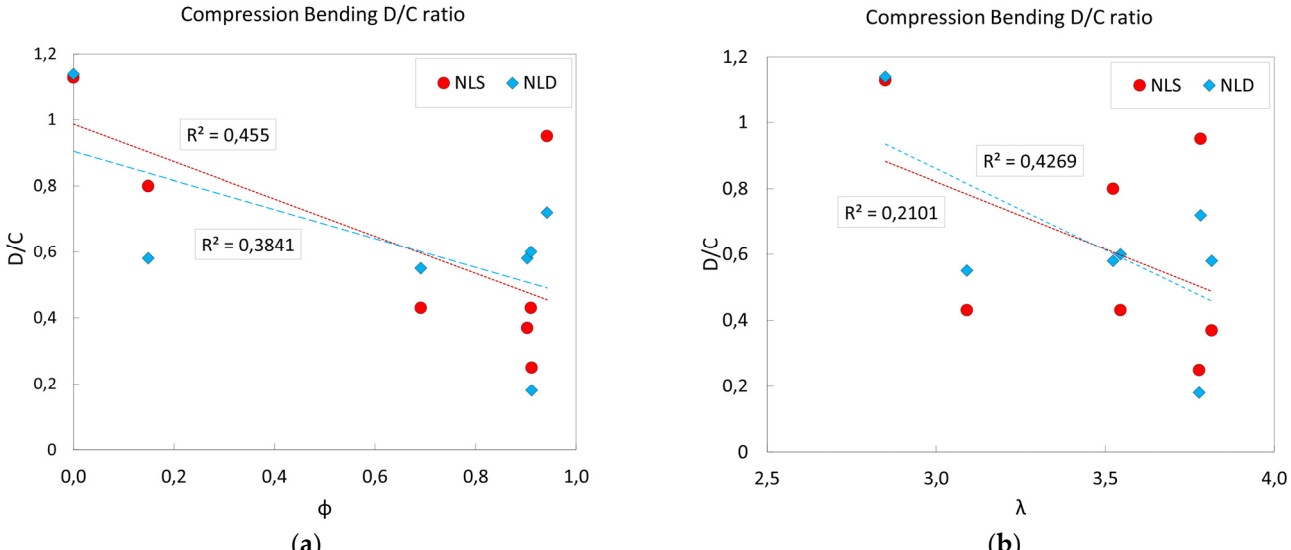

**Figure 11.** Compression–bending failure distribution for a standard section as a function of the filling ratio φ (**a**) and slenderness λ (**b**). Results of the NLS and NLD analyses.

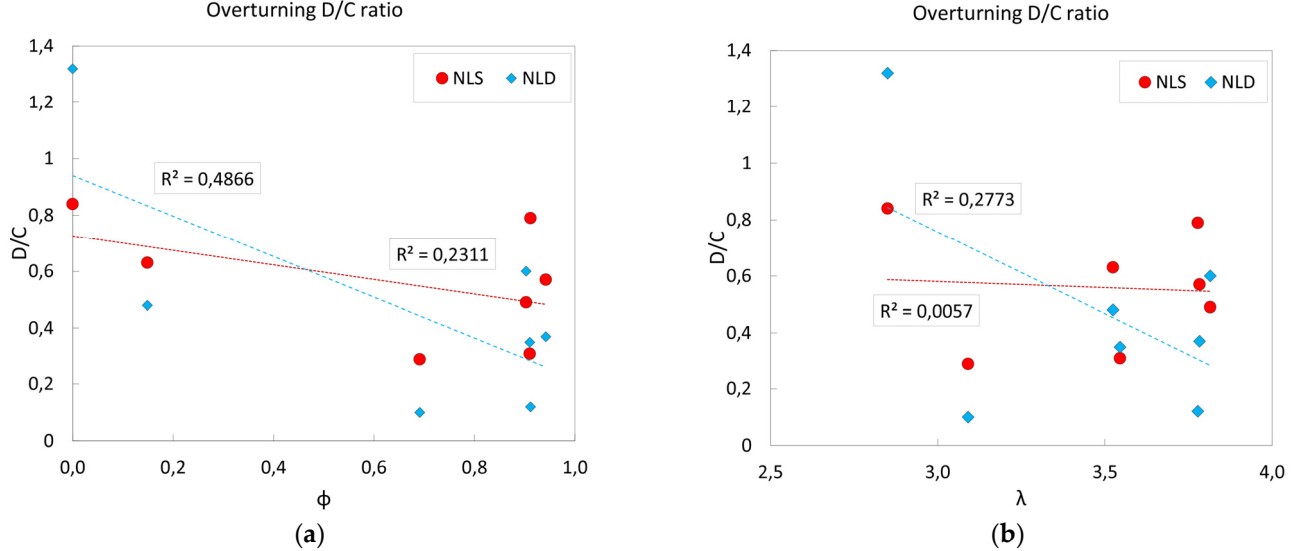

**Figure 12.** Overturning distribution for a standard section as a function of the filling ratio φ (**a**) and slenderness λ (**b**). Results of the NLS and NLD analyses.

## 5. Discussion

The in situ survey and the occurring failures showed that degradation elements such as moisture, vegetation, and presence of buildings behind the wall can represent significant exposure elements for a HUW.

As shown by the NLS analysis (Figure 8), considering imbibition results in a significant difference in structural response. Table 5 highlights the difference in terms of the ultimate displacement ratio Δμ, which ranged between −0.23 and 5.83.

The static and dynamic analyses carried out on sections with different slenderness and filling ratios showed that the D/C ratio is a significant indicator of the ultimate LS behavior of the section (Figures 11 and 12). The analysis also highlighted that the filling ratio had a better correlation with D/C than slenderness, despite the universal importance of λ in the mechanical characterization of masonry walls. This was especially evident in the dry scenario and reflects the large impact of soil interactions on the analysis of a retaining wall.

The capacity and the type of failure/mechanism (Figure 9) can be significantly affected by the presence of moisture. For the scenario with imbibition, the shear failure became the most probable. The displacement limit, assumed as a percentage of the height of the wall, can be used as a significant indicator of vulnerability.

Imbibition also affects the shape and dimensions of the failure surfaces (Figure 10). This is a significant factor in a case where the HUW has a close proximity with the city, as in the case of Volterra, since the imbibition affects the failure surface and, involving a higher collapse area, could result in damage to people and to historical buildings or artefacts.

The comparison between the results of the different kinds of analysis run on the plain model of the HUW sections, in terms of the failure surfaces and failure type (Figures 10–12), at this moment, does not allow determining if a certain kind of analysis is more safe than the others. It would be necessary to extend the study to a wider case series to confirm or disprove these considerations.

## 6. Conclusions

In this paper the role of soil imbibition in the seismic vulnerability analysis of a HUW was analyzed, considering the SLIP model as a model of imbibition. Structural analyses were carried out with plane models on some relevant sections of the HUW of Volterra (Italy), which were characterized by different filling ratios $\varphi$. The analyses were performed with both LEM (SSAP) and with FEM (Straus7), as well as carrying out both NLS and NLD analyses. The results indicated that the presence of soil moisture plays a significant role in terms of the structural safety. The demand/capacity ratio is highly affected by moisture. Compression–bending failure appeared to be the most probable mechanism in the dry scenario, whereas the presence of moisture made shear failure more probable.

The traditional and codified approach to carrying out an analysis in dry or undrained conditions, although generally accepted, has some limits in the case of a structure with extensive soil interaction. This present paper aimed to improve on this traditional approach, in which the soil is usually considered in a conventional way, in undrained conditions, and where the presence of accidental water due to rainfall, uncontrolled seepage motions, and water losses, which can significantly aggravate the static and seismic behavior, is neglected.

More in depth studies are needed to identify a standard procedure for considering these effects. It is strongly recommended to consider moisture effects in the safety analysis in all cases of strong structure–soil interaction, not only as an additional load (hydraulic thrust), but also as a reduction in the mechanical parameters of the soil.

**Author Contributions:** Conceptualization and methodology, M.S.; Investigation and resources, M.D.; writing—review and editing, G.C.; funding acquisition, M.S. All authors have read and agreed to the published version of the manuscript.

**Funding:** This research was funded by Italian Ministry of University Research—project PRIN 2020 S-MoSES.

**Data Availability Statement:** Not applicable.

**Acknowledgments:** Thanks to the Italian Ministry of University Research for financing the S-MoSES Project with PRIN 2020 programme.

**Conflicts of Interest:** The authors declare no conflict of interest.

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
