# Peer review of "Seismic Analysis of Historical Urban Walls: Application to the Volterra Case Study"

_infrastructures, doi:10.3390/infrastructures8020018_

Round 1

Reviewer 1 Report

(1) The paper reports the seismic assessment of historical urban walls, considering the Volterra walls as case study. The seismic response of several wall sections (also including retaining walls) is investigated by means of both nonlinear static and dynamic analyses, carried out considering both limit equilibrium analysis and finite element method, and considering four limit state conditions. The study characterizes the influence of imbibition on the seismic performance of the investigated walls. Particular focus is on the identification of the failure mechanisms, as well as on the influence of the key parameters on the performance measures (i.e., demand to capacity ratios).

(2) The manuscript is well-organized and well-written. English and technical terminology are appropriate. The Introduction presents and discusses well the technical and scientific background, as well as the relevant literature studies are reported; further studies focused on the seismic assessment of rocking-dominated elements, such as the case study walls, might be added to enrich the literature review. The methodology is well described. The results are presented in an efficient manner, as well as the discussion part is effective and based on the provided evidence. The conclusions are supported by the data.

(3) The paper potentially contributes to the technical and scientific literature since the seismic response of the investigated historical walls is characterized considering several key parameters, and highlighting the key influence of the imbibition on the seismic performance. In this sense, the manuscript develops novel knowledge and provides applicative tools/means for carrying out seismic vulnerability assessment of similar structures. However, minor revisions should be implemented by the Authors in order to improve the overall quality of the paper, in order to make the manuscript suitable for publications. Please, find the detailed review comments in the attached reviewed manuscript report.

Author Response

Dear reviewer,

warm thanks for your comments! They permitted us to improve the manuscript. We have followed all the comments with care, it is possible to find in the revised version. Particularly we have rewritten abstract and introduction, he have enforced the references, we have improved all the images and tables, verifying also english fluency. Finally we have better pointed out the novelty of the manuscript respect to the State-of-Art

Reviewer 2 Report

The manuscript entitled “Seismic analysis of historical urban walls: application to the Volterra case study” investigated  seismic  assessment of selected historical walls.

After careful review, the manuscript has a reasonable effort and technical information. Firstly, I am not sure about the validation procedure and how is the applicability and acceptance of the method confirmed and compared. What about soil and site conditions? It has to be clarified in the next response letter. However, there is not much novelty on the work as there are many similar works, and there are some points that must be considered in the revision to be worth being accepted. Therefore, I strongly recommend the authors to follow the comments below:

1- Abstract does not present important points. It is confusing a bit too. Why thecollapse of historical urban walls is a critical issue in Civil Engineering? The abstract should be between 250 to 300 words and concisely mention the problems of previous works and novelties in this paper.  

Similarly, the introduction has very poor structure and lack of literature review. Usually in the chapter of introduction the background and needs of this study and why it has to be highlighted and prepare readers to go further. Then your second chapter should be literature review where you present an overview on the previous works and the main problem statements of work and how it can be improved or overcome on it. You can concise it and point the important parts and highlights the achievements.

2- In many parts of the paper your reference to the figure or table and reference list was failed and there are errors that has to be fixed.

3- The work presents a poor literature review on the methods based on modern techniqes such as soft computing (ML, Fuzzy and AI) for seismic vulnerability and fragility/damage assessment of historical and existing buildings/walls/bridges. Because your work is not much new and novel, I strongly recommend you make your introduction and study on previous methods interesting for readers by adding the following new works which I found it new and related to your work which are based on different methods for vulnerability and damage assessment of buildings and infrastructures. It will increase the depth of your review and stronger and wider area.

-Structural analysis of five historical minarets in Bitlis (Turkey)

-SLaMA-URM method for the seismic vulnerability assessment of UnReinforced Masonry structures: Formulation and validation for a substructure

-Seismic Vulnerability Assessment of Old Brick Masonry Buildings: A Case Study of Dhulikhel

-Seismic fragility analysis of masonry structures considering the effect of mainshock-aftershock sequences

There are also many other methods based on ML and other methods that can be found on the internet to add.

4- All Figures: Please improve the quality and font size to be more visible. Please make the graphic nice and well cropped. Otherwise if you have used it from somewhere else please cite it otherwise is plagiarism.

Size of figures, fonts and legend must have harmony, please revise it well. 

5- Please provide a figure that shows the architecture of your model and work progress. Something like a graphical abstract.

6- Please make your tables and figures follow a same path and font size and colors and adjust them in a proper way.

7- There are many typos that needs to be corrected and please make the format similar for all. Also, be careful about spaces after (.) amd (,) which in some cases are doubled or missed.

8- Generally, provide more information about the data.  The repository of data collections and it would be helpful to provide some technical information about the possible earthquake that can happen or happened on the case study area such as magnitude, PGA and other related information.

9- You did not highlight the problem statement, objectives and novelty of your proposed method; That is why increasing the background of literature review based on the recommended works can help in this manner.

10- There is a need in proofreading the work.

11- Please provide more information about the figures and tables and write more in the body of the text about them and the information they provide.

At the end as I have mentioned, there are not much significant novelty on this work but the efforts were good and it would be good if you revise it according to the points provided and other reviewers.

Author Response

(The authors gave the same response as above.)

Reviewer 3 Report

A) General remarks
The research presents in this paper a very interesting topic, as well as results that are of wider significance when it comes to seismic performance assessment, especially for historical civil engineering structures. The paper is concise and clear. The literature in the paper is adequately cited, however, some comments on the choice and significance of cited sources will be articulated in the points below.
1.     In the case of literature, all the cited references are relevant to the research. No unnecessary self-citations were detected.
2.    The English language used is without major mistakes and no significant connections are needed.
3.    The abstract is well written. The role of the abstract is to give a basic overview of the paper. In this case, the abstract gives a good introduction to the paper without specific data and is very informative even for those none-familiar with the topic reader.  However, the novelty aspects of the paper are not presented.
4.    The introduction is mostly well-written and follows all the rules of the proper instruction on the topic. However,
       a.    Some paragraphs present only basic information that suits the authors the best but does not present fully state of the art (look at example point c).
      b.    The first sentence- “recently” – 3 years ago is not really recent. Please rephrase. This whole paragraph needs to be improved cause it is a little bit confusing to read.
      c.    The paragraph about civil engineering monitoring (lines 49-52 )is very basic and does not include typical methods (seismic stations) or monitoring over a long period of time and data analysis like probabilistic power spectral density technique (e.g. DOI 10.21008/j.0860-6897.2020.3.11). Please improve this paragraph.
      d.    Additionally, at the end of the introduction, the aim, scope and novelty of the proposed paper are not fully introduced.  The novelty aspect is not really presented.

5.    The biggest problem of the article is the weak statement on the novelty of the paper. How the presented research is different from other papers in the field. What new elements are introduced? Because this is a case study type of article this must be clearly stated. Without this, the paper does not bring much in case of new knowledge and does not meet the criterium to be published as a scientific paper. The biggest problem of the whole article is the statement of novelty. This must be presented strongly. Without it is mostly the presentation of engineering tools for specific engineering case analysis.
6.    There is a problem with references. The error “Error! Reference source not found” occurs. This may be a problem with how the document was converted to pdf. Please check before submitting the future version.
7.    The methodology and result presentation are clear. No major problems were detected.
8.    The paper does provide proper conclusions. However, again the novelty aspect must be pointed out strongly.

B) Item remarks
Fig .2 the section numbers are not really visible.
Fig. 8 is very important but not all information is visible/readable. This is especially in the case of legends and horizontal load values.
Fig. 10 legend texts not readable.

C) Conclusions:
The biggest problem of the article is also the clear presentation of the novelty of the research topic. In its current form, without a novelty component, it is at most a borderline paper which currently does not meet the criterium for publication in a scientific journal.
The reviewer suggests the major corrections mentioned previously and asks the authors to answer the fundamental questions- how the article can advance this field of study?

Author Response

Dear reviewer,

warm thanks for your comments! They permitted us to improve the manuscript. We have followed all the comments with care, taking into account into the revised version where it is possible to find it. Particularly we have rewritten abstract and introduction, he have enforced the references, we have improved all the images and tables, verifying also english fluency. Finally we have better pointed out the novelty of the manuscript respect to the State-of-Art.

Round 2

Reviewer 2 Report

Dear Authors

Many thanks for your response and significant changes.

Reviewer 3 Report

Dear authors,

The reviewer has noticed significant improvements.

However, some aspects and the quality of the paper still can be improved, if no additional elements are asked to be improved by the other reviewer, the paper can be considered for eventual publication.

Best regards,

The reviewer